# Simultaneous Measurement of 6DOF Motion Errors of Linear Guides of CNC Machine Tools Using Different Modes

**DOI:** 10.3390/s20123439

**Published:** 2020-06-18

**Authors:** Peizhi Jia, Bin Zhang, Qibo Feng, Fajia Zheng

**Affiliations:** Key Lab of Luminescence and Optical Information, Ministry of Education, Beijing Jiaotong University, Beijing 100044, China; 18118030@bjtu.edu.cn (P.J.); qbfeng@bjtu.edu.cn (Q.F.); 16118441@bjtu.edu.cn (F.Z.)

**Keywords:** 6DOF motion error, linear axis, measurement model, measurement mode, machine tool

## Abstract

Based on the prior work on the six degrees of freedom (6DOF) motion errors measurement system for linear axes, and for the different types of machine tools and different installation methods, this study used a ray tracing idea to establish the measurement models for two different measurement modes: (1) the measurement head is fixed and the target mirror moves and (2) the target mirror is fixed and the measurement head moves. Several experiments were performed on the same linear guide using two different measurement modes. The comparative experiments show that the two measurement modes and their corresponding measurement models are correct and effective. In the actual measurement process, it is therefore possible to select the corresponding measurement model according to the measurement mode. Furthermore, the correct motion error evaluation results can be obtained.

## 1. Introduction

Linear guides are an important part of precision machining equipment, such as computer numerical control (CNC) machine tools. Linear guides have motion errors that directly affect the machining accuracy. The error compensation method can effectively improve the accuracy, and this multiparameter, high-precision, and fast measurement method is the key to error compensation [1,2]. The measurement method based on laser interference is the conventional one [3,4,5,6]. Relevant commercial interferometers produced by Renishaw, Keysight and JENAer are mostly used for single parameter measurement. With the increasing requirements for measurement efficiency, multiparameter simultaneous measurement systems based on the principle of laser collimation [7,8,9,10] or the combination of collimation and interference [11,12,13,14,15,16,17,18] continue to emerge, and have become the current research trend. Although, multiparameter simultaneous measurement systems have improved efficiency compared to single-parameter measurement systems, in actual measurement, it is often necessary to adjust the installation positions of the measurement head and the target mirror according to the machine tool types. This complicates the installation and commissioning process of the measurement system. Therefore, it is meaningful to study a uniform and convenient installation method, which can meet the measurement of linear guides of all types of machine tools.

Generally, there are two installation methods for the measurement system, such as mounting the measurement head on the tripod and the target mirror on the worktable or spindle [17]. Another installation method is to install the measurement head and the target mirror on the worktable and the spindle, respectively. When performing measurements, it is essential that the machine tool has the same impact on the vibration of the measurement head and the target mirror. Otherwise, it may cause measurement error. Simultaneously, it must meet the measurement requirements of various types of CNC machine tool linear guides and it should simplify the installation and adjustment process. Therefore, the second installation method is currently used more. For example, Renishaw’s XM60 laser interferometer [19] and API’s XD laser interferometer [20] both use this installation method. However, based on this installation method, there are two different measurement modes in the measurement process due to the different types of machine tools. In general, the default measurement mode is the fixed measurement head, and the target mirror moves with the worktable or spindle [9,10,13,17,18,21]. This measurement mode is called measurement mode 1 in this paper. However, due to the installation method and the machine tool types, there is also a measurement mode in which the target mirror is fixed and the measurement head moves. This measurement mode is called measurement mode 2 in this paper. Table 1 lists the measurement modes of the linear guides of the different types of CNC machine tools when the measurement head and the target mirror are installed on the worktable and the spindle, respectively. This shows that the installation method of the measurement system and the type of machine tools determine the measurement mode. Generally, during the measurement process, the default measurement mode is measurement mode 1, and measurement mode 2 is ignored. However, if measurement mode 2 can also be used to measure the motion errors, then “install the measurement head and the target mirror on the worktable and the spindle, respectively” can be used as the unified installation method, regardless of the machine tool types. The measurement process of the four types of machine tools after the unified installation method is shown in Figure 1. Therefore, the installation method of the multiparameter simultaneous measurement system is unified, the installation and adjustment process of the measurement system is simplified, and the universality of the measurement system is improved. Furthermore, based on this unified installation method and the two measurement modes, if the motion errors of three linear axes can be measured automatically in one installation, the measurement efficiency will be significantly improved. However, so far, there is no detailed theoretical analysis to prove that the error measurement in measurement mode 2 can obtain the correct evaluation results. Therefore, this article focuses on the inherent differences between the two measurements modes and uses an independently developed 6DOF motion errors measurement system based on the principles of interference and laser collimation to thoroughly analyze the two measurement modes.

In summary, based on the linear axis motion errors measurement system developed by our research group, this paper analyzes the differences between the linear axis 6DOF motion errors measurement models for the two measurement modes. The structure and measurement principles of the measurement system are described in detail in Section 2. Based on the ray tracing and matrix analysis methods, the motion error measurement models are established, and the differences between the two different measurement modes are analyzed. This is further analyzed in detail in Section 3. In Section 4, the experimental results are shown to prove the accuracy and effectiveness of the measurement modes and the corresponding models which are proposed in this paper. Finally, Section 5 delivers the conclusion of this paper.

## 2. A Simultaneous Measurement System and the Principle of a 6 DOF Motion Error for a Linear Axis

The measurement system for simultaneously measuring the 6DOF motion errors of a linear axis is based on the principles of laser interference and laser collimation. Figure 2 shows the configuration of the self-made measurement system [22,23]. This consists of a laser–fiber coupling unit, a measurement head, and a target mirror. The dual-frequency laser generated by the He-Ne laser is coupled into a single-mode polarization maintaining fiber (SMF) as the light source of the measurement system. At the other end of the fiber, the collimated measurement beam is divided into two beams by the beam splitter ($BS_{1}$). One is used as the reference beam for the heterodyne interference measurement, and the other is used as a measurement beam. The transmitted and reflected beams from the polarization beam splitter ($PBS_{1}$) are reflected by the corner cube reflectors ($RR_{3}$,$RR_{1}$), respectively. The two reflected beams are combined and irradiated onto the detector ($D_{2}$) after the polarization beam splitter ($PBS_{1}$) and a polarizer ($P_{1}$). This is the measurement signal of the positioning error ($\delta_{z}$). The reflected beam of the corner cube reflector ($RR_{3}$) is received by the detector ($QD_{2}$) after being reflected by the beam splitter ($BS_{3}$). In addition, the reflected beam of the corner cube reflector ($RR_{2}$) is received by the detector ($QD_{1}$). The two straightness errors and the roll angle error are calculated based on the data obtained by these two detectors. The reflected beam of beam splitter ($BS_{4}$) is used to measure the yaw and pitch of the linear axis after the polarization beam splitter ($PBS_{2}$), a mirror ($M_{2}$), and a lens (L).

The straightness errors $\delta_{x}$ and $\delta_{y}$ are measured based on the principle of laser collimation. The spot on $QD_{1}$ will have relative displacements ($\Delta X_{QD1}$,$\Delta Y_{QD1}$) when there are horizontal and vertical displacements ($\delta_{x},\delta_{y}$). In addition, the spot on $QD_{2}$ will produce the relative displacements ($\Delta X_{QD2}$,$\Delta Y_{QD2}$). Therefore, the horizontal and vertical straightness errors of a linear guide can be calculated based on the relative position changes of the spots on the detectors $QD_{1}$ and$\;QD_{2}$.

The yaw (α) and pitch (β) are measured based on the principle of autocollimation. The spot on the detector (PSD) will simultaneously generate a relative position change in the horizontal and vertical directions ($\Delta X_{PSD}$,$\Delta Y_{PSD}$) because of these two angles. The pitch and yaw of the linear guide can be calculated. Because the focusing lens (L) is added to the measurement system, the spot on the detector is not affected by the crosstalk of the straightness errors.

The roll error (γ) is measured based on the principle of laser collimation. The light spots on the detectors $QD_{1}$ and $QD_{2}$ will produce relative displacements $\Delta Y_{QD1}$ and $\Delta Y_{QD2}$, respectively, in the vertical direction owing to the roll. According to the relative position change of $\Delta Y_{QD1}$, $\Delta Y_{QD2}$ and the distance (h) between the two measuring beams, the roll of a linear guide can be calculated.

## 3. Measurement Models in Two Measurement Modes

During the measurement process, the target mirror and the measurement head are fixed on the machine tool’s spindle and workbench, respectively. Due to the different types of machines, there are two different measurement modes. In measurement mode 1, the measurement head is fixed, and the target mirror moves in the direction of the axis to be measured. In measurement mode 2, the target mirror is fixed and the measurement head moves in the direction of the axis to be measured. In order to analyze the differences between these two measurement modes, linear guide motion error measurement models were established. The principle of straightness and the angle error measurement for the two different measurement modes is illustrated in Figure 3.

The positioning error of the linear axis is measured by a dual-frequency laser interferometer; this measurement model is not discussed here. Other five degrees of freedom (5DOF) motion errors are based on the idea of ray tracing [24,25]. A Cartesian coordinate system for the measurement head and target mirror was established. The coordinate system conversion matrix is calculated according to the actual manufacturing parameters and the relative position relationship of the measurement head and the target mirror. The surface of each optical device in the measurement head and the two corner cube reflectors is abstracted as a space plane. In addition, the measurement beam is abstracted as a straight line in space. The beam is refracted or reflected on the surface as it passes through the optical planes. The plane’s refraction matrix and reflection matrix are established to analyze the influence of each plane on the beam propagation direction. The beam carrying the motion errors of a linear guide passes through the corner cube reflectors or plane mirror and it returns to the detectors. The intersection point coordinates of the light beam and each optical plane, and the direction of beam propagation after refraction and reflection are analyzed one by one. Then, the intersection point coordinates of the light beam and the detector are obtained. The straightness and angular error measurement models can be obtained by the expression of the coordinates of the intersection of the light and the detector. The coordinate system is established as demonstrated in Figure 4. X1Y1Z1 and X2Y2Z2 are the coordinates of the measurement head and the target mirror, respectively. In order to facilitate the calculation, a model was established using MATLAB. The geometric parameters of the optical device, such as the corner cube reflector and the mechanical parameters of the target, are introduced into the error model. The simplified function of MATLAB was used to obtain the final measurement model.

### 3.1. Measurement Mode 1

For measurement mode 1, the measurement head is fixed and the target mirror moves in the direction of the axis to be measured. According to the previous modeling method, the measurement model is shown in Equations (1)–(5).

Equations (1) and (2) are for the yaw (α) and pitch (β), respectively, where $\Delta X_{PSD}$ and $\Delta Y_{PSD}$ are the relative position changes of the spot on the PSD in the horizontal and vertical directions, respectively. In addition, f is the focal length of the lens. Because the lens (L) is used, the measurement of the yaw and pitch are not affected by the straightness error.
(1)$$\alpha =\frac{\Delta X_{PSD}}{2f}$$
(2)$$\beta =\frac{\Delta Y_{PSD}}{2f}$$

Equations (3) and (4) are for the straightness error, where $\Delta X_{QD1}$, $\Delta Y_{QD1}$, $\Delta X_{QD2}$ and $\Delta Y_{QD2}$ are the relative position changes of the spot on $QD_{1}$ and $QD_{2}$, respectively. The yaw and pitch will cause an intersection of the measurement beam and the bottom surface of the corner cube reflector. This will indirectly affect the change of the spot position on the detectors $QD_{1}$ and $QD_{2}$, and ultimately affect the accuracy of the straightness error measurement. The side length of the equilateral triangle of the incident surface of the corner cube reflector is represented by a. The refractive index of the corner cube reflector is represented by n.
(3)$$\delta_{x}=\frac{\left(\Delta X_{QD1}+\Delta X_{QD2}\right)\pm \frac{2\sqrt{6}\times a\times \alpha }{3\times n}}{4}$$
(4)$$\delta_{y}=\frac{\left(\Delta Y_{QD1}+\Delta Y_{QD2}\right)\pm \frac{2\sqrt{6}\times a\times \beta }{3\times n}}{4}$$

The roll angle error measurement is calculated based on the vertical variation of the spot on the two QD detectors. According to Equation (5), it is known that the influence of the error crosstalk on $\Delta Y_{QD1}$ and $\Delta Y_{QD2}$ is the same. In addition, it can be offset during the difference calculation.
(5)$$\gamma =\frac{\Delta Y_{QD1}-\Delta Y_{QD2}}{2h}$$

### 3.2. Measurement Mode 2

For measurement mode 2, the target mirror is fixed and the measurement head moves in the direction of the axis to be measured. The measurement model is presented in Equations (6)–(10), according to the previous modeling method.

The angle error measurement model featured in Equations (6) and (7) is the same as measurement mode 1.
(6)$$\alpha =\frac{\Delta X_{PSD}}{2f}$$
(7)$$\beta =\frac{\Delta Y_{PSD}}{2f}$$

The straightness error measurement models are shown in Equations (8) and (9). Because the measurement beam is affected by the yaw and pitch of the linear axis, the beam incident on the bottom surface of the corner cube reflector and the reflected beam have an angular deviation in space. This angular deviation and the measurement distance work together, which leads to angular error crosstalk for the straightness measurement. This error crosstalk also includes the effect of the beam refraction in a corner cube reflector. The distance between the measurement head and the target mirror is represented by D.
(8)$$\delta_{x}=\frac{\left(\Delta X_{QD1}+\Delta X_{QD2}\right)\pm \left(\frac{2\sqrt{6}\times a\times \alpha }{3\times n}+4*\alpha *D\right)}{4}$$
(9)$$\delta_{y}=\frac{\left(\Delta Y_{QD1}+\Delta Y_{QD2}\right)\pm \left(\frac{2\sqrt{6}\times a\times \beta }{3\times n}+4*\beta *D\right)}{4}$$

Equation (10) is for the roll error in mode 2. The roll measurement models are the same for the two measurement modes.
(10)$$\gamma =\frac{\Delta Y_{QD1}-\Delta Y_{QD2}}{2h}$$

In summary, the difference between the two measurement modes is primarily the straightness error measurement model. Obviously, the difference term of the straightness error measurement model is α(β)×D. This difference term is not a slope error. α(β) is the yaw or pitch of the measurement head at a sampling point of the linear guide, and the angular error at each sampling point is different. It is not the angle between the measuring beam and the moving direction of the linear guide. Therefore, the difference term α(β)∗D can be regarded as the angular error crosstalk particular to measurement mode 2. This crosstalk term is also related to the measurement distances. The larger the angular error of the rail to be measured, the farther the measurement distance is and the greater the influence of the error crosstalk on the accuracy of the straightness error measurement is; therefore, it cannot be ignored. In the actual measurement process, different measurement modes need to be matched with different measurement models to ensure the measurement accuracy of the straightness errors for the two measurement modes.

## 4. Experimental Results and Analysis

In order to prove the feasibility and effectiveness of the two measurement modes and the matched measurement models, the comparative experiments were conducted on the same linear guide for measurement modes 1 and 2, respectively. The experimental process is demonstrated in Figure 5—Figure 5a is for measurement mode 1 and Figure 5b is for measurement mode 2. The linear axis used in this experiment is the ABL2000 series. It features an air bearing that is directly driven by a linear platform, which was produced by AEROTECH. The air float guide has an accuracy of ± 0.75 µm.

Based on the analysis in the previous section, the main difference between the two measurement modes is the straightness error measurement model. Therefore, in this section, an indepth analysis of the differences in straightness error measurement models is performed.

### 4.1. Performance of 6DOF Error Measurement System

In order to demonstrate the performance of the 6DOF motion errors measurement system, in the previous works, the calibration, the stability, the repeatability and the resolution were carried out based on measurement mode 1. The above experiment was conducted under laboratory conditions, with an air temperature range of about 24 ± 0.2 °C, a relative humidity of 23.5 ± 1% and a pressure of 1014.5 mbar.

A grating ruler (LG-50, accuracy: 0.1 μm, resolution: 50 nm) was used to calibrate the straightness errors, and a photoelectric collimator (Collapex EXP, accuracy: 0.2 arcsec, resolution: 0.01 arcsec) was used to calibrate the yaw and pitch. The results show that the linear fitting determination coefficient can reach 0.9997 for the straightness in the measurement range of ± 100 μm, and the linear fitting determination coefficient can reach 1 for pitch and yaw in the measurement range of ± 200 arcsec. According to the roll measurement model and the measurement range of straightness, the measurement range of the roll angle can be calculated to be ± 680 arcsec. The measurement range of the positioning error is affected by the phase demodulator (E1735A USB Axis Module, Keysight, Beijing, China) and the quality of the measurement beam, which is about 5 m.

The stability experiment was carried out within 30 min. The standard deviation of stability is listed in the “Stability” column of Table 2, which proves that the proposed system has good stability. Three measurements were carried out for the same linear guide, and the formula “(Maximum-Minimum)/2” was used to evaluate the repeatability of the measurement system. The results of the repeatability error are shown in Table 2.

Based on the measurement models for pitch, yaw and straightness errors, the resolution of the detector QD and PSD and the focal length of the lens, the error measurement resolution of the pitch, yaw and the straightness can be calculated. The measurement resolution of the roll can be calculated by the measurement model and the resolution of QD detectors. The measurement of positioning error is based on the principle of heterodyne interferometer, and its measurement resolution depends on the performance of the phase demodulator (E1735A USB axis module). Table 2 shows the results for all resolutions.

Table 2 describes the detailed parameters of the self-made 6DOF motion errors measurement system. The experimental results show the feasibility and accuracy of the measurement system.

System errors usually affect the performance of the measurement system, including the manufacturing and installation errors of optical components, the angle drift of the measurement beam, and the installation errors of the detector in the measurement head. These errors can be compensated by adding a common-path beam drift measurement and compensation structure [25] and establishing an error compensation model [24,26].

### 4.2. Analysis of the Straightness Error Measurement Models

According to the analysis results in Section 3, only the straightness error measurement model was different for the two different modes. This difference is usually ignored. Using the measurement model corresponding to mode 1, namely Equations (3) and (4), the calculation of the straightness is obtained under mode 2. Taking measurement mode 1 as the standard, we analyze measurement mode 2. The results are shown in Figure 6. The comparison results show that the maximum contrast deviations of horizontal straightness and vertical straightness were 5.76 and 7.69 μm, respectively. It can be seen that the differences between the two modes cannot be ignored. Otherwise, straightness evaluation results with large errors will be obtained.

### 4.3. Comparative Experiment between Two Measurement Modes

Based on this 6DOF error measurement system, three measurements were performed on the same linear guide using two measurement modes within 10 min, and the average of the measurement results was taken for comparison. The measurement distance was 500 mm and the measurement interval was 50 mm. According to the different measurement modes, corresponding measurement models are used to evaluate the motion errors. The positioning error, straightness, pitch and yaw measurements were compared with a laser interferometer (XL-80, Renishaw, linear resolution: 1 nm, angular resolution: 0.01 arcsec). The comparison measurement for the roll was conducted using an electronic level (WL11, Qianshao, accuracy: 0.2 arcsec). The comparison results of the two measurement modes and the commercial instruments are shown in Figure 7.

The comparison results of the two measurement modes are shown below. The maximum comparison deviation of the positioning error was 0.16 μm; the maximum comparison deviation of the horizontal straightness and vertical straightness are 0.57 and 0.54 μm, respectively. The maximum comparison deviations of the yaw and pitch are 0.48 and 0.62 arcsec, respectively; the maximum comparison deviation of the roll angle is 1.58 arcsec.

The comparison results between the commercial instruments and our system in measurement mode 1 are shown below. The maximum comparison deviation of the positioning error was 0.27 μm; the maximum comparison deviations of the horizontal straightness and vertical straightness were 0.49 and 0.84 μm, respectively. The maximum comparison deviations of the yaw and pitch were 0.65 and 0.51 arcsec, respectively; the maximum comparison deviation of the roll angle was 2.11 arcsec.

The main reason for the comparison deviations is that our measurement instruments under two measurement modes and the commercial instrument cannot accurately measure the same point on the linear guide.

The experimental results show that no matter which measurement mode is used, a unified measurement model can be used to evaluate angle errors, and there is no difference between their measurement models. Although there are differences between the straightness error measurement modes, the correct error evaluation result can be obtained by using the measurement model corresponding to the measurement mode.

## 5. Conclusions

This paper analyzes the simultaneous measurement model of the 6DOF motion errors of the linear guide of a CNC machine tool under two different measurement modes. The results show that under two different measurement modes, the straightness error measurement is significantly different, and the positioning error and angle error measurement models are the same. Therefore, in order to ensure the measurement accuracy of the straightness errors in two different measurement modes, different measurement models need to be selected according to the different measurement modes. The experimental results of the comparison between the two measurement modes can prove that the two measurement modes and the corresponding measurement models are correct and effective. The two measurement modes and measurement models proposed in this paper can meet the requirements for simultaneous measurement of the motion errors of various types of CNC machine tool linear guides, thereby improving the universality of the existing measurement systems. Based on the research results of this paper, the high-precision and high-efficiency measurement methods of multi-axis CNC machine tools will be further studied.

## Figures and Tables

**Figure 1 sensors-20-03439-f001:**
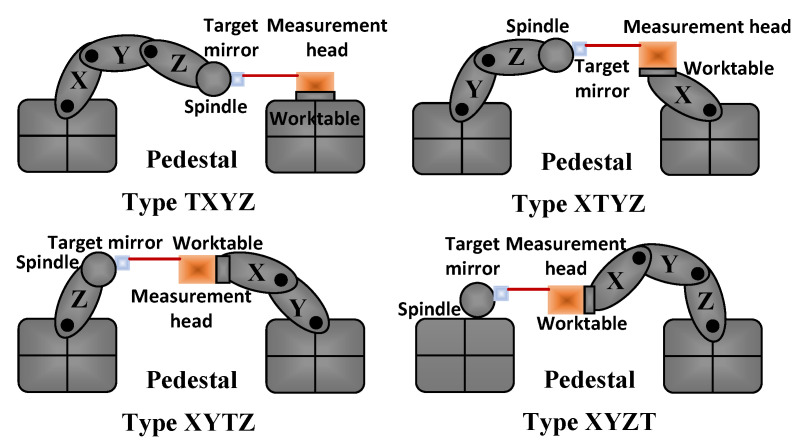
Measurement process of the different types of machine tools with unified installation.

**Figure 2 sensors-20-03439-f002:**
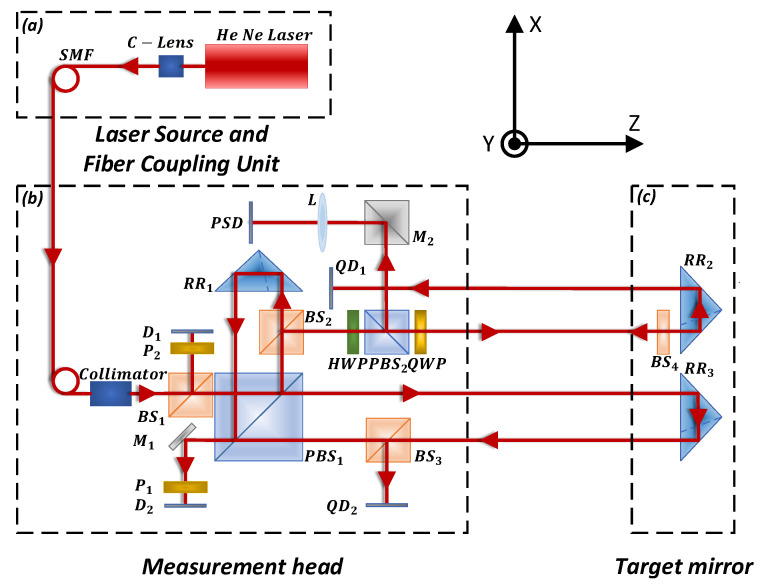
Configuration of the simultaneous measurement system for 6DOF errors on a linear axis.

**Figure 3 sensors-20-03439-f003:**
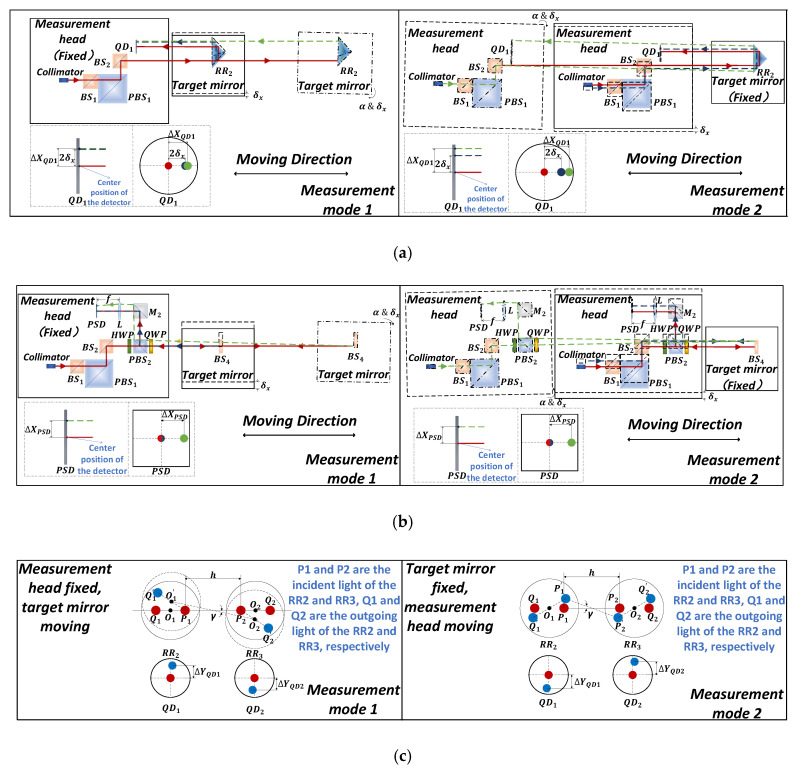
Principles of straightness and angular error measurement under two different measurement modes. (**a**) Straightness error measurement; (**b**) Yaw (pitch) error measurement; (**c**) Roll error measurement.

**Figure 4 sensors-20-03439-f004:**
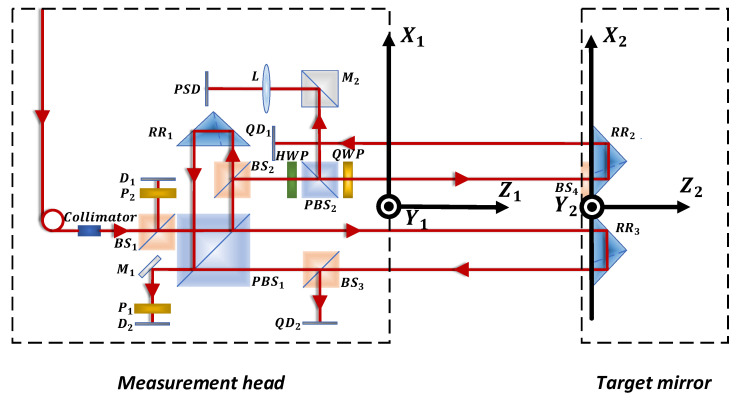
Coordinate system for the measurement head and target mirror.

**Figure 5 sensors-20-03439-f005:**
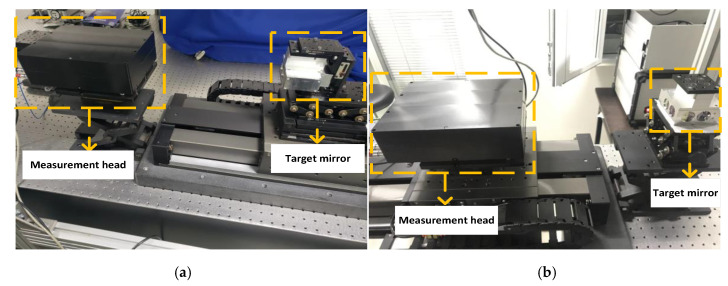
Linear axis motion errors measurement experiment for the two measurement modes. (**a**) Measurement mode 1; (**b**) Measurement mode 2.

**Figure 6 sensors-20-03439-f006:**
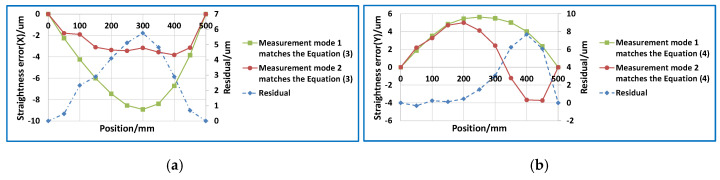
Differences between straightness error measurement models under two measurement modes. (**a**) Straightness error (X); (**b**) Straightness error (Y).

**Figure 7 sensors-20-03439-f007:**
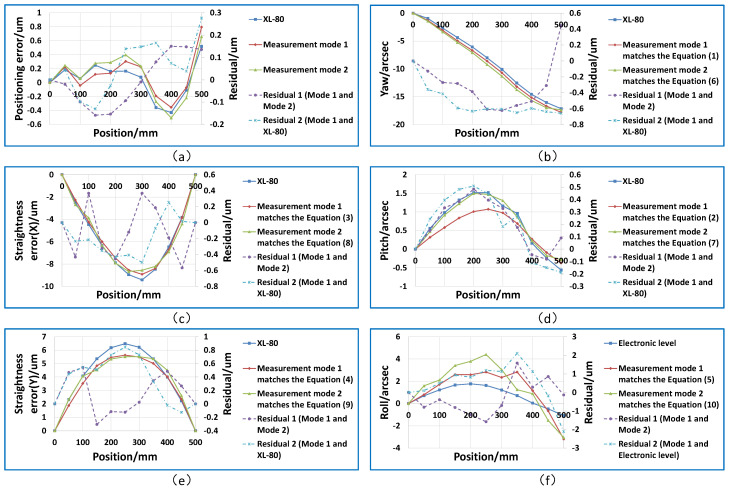
Experimental results of comparison between the two measurement modes and between measurement mode 1 and commercial instruments. (**a**) Positioning error; (**b**) Yaw error; (**c**) Straightness error (X); (**d**) Pitch error; (**e**) Straightness error (Y); (**f**) Roll error.

**Table 1 sensors-20-03439-t001:** Linear axis measurement modes of the different types of machine tools.

CNC Machine Type	Measurement Mode 1	Measurement Mode 2
Type TXYZ	Axis X, Y, Z	None
Type XTYZ	Axis Y, Z	Axis X
Type XYTZ	Axis Z	Axis X, Y
Type XYZT	None	Axis X, Y, Z

**Table 2 sensors-20-03439-t002:** Parameters of 6DOF motion errors measurement system.

Parameter	Stability Standard Deviation	Repeatability Error	Resolution	Measurement Range
Positioning error	40 nm	±30 nm	1 nm	5 m
Straightness error (X)	0.07 μm	±0.25 μm	0.1 μm	±100 μm
Straightness error (Y)	0.09 μm	±0.37 μm	0.1 μm	±100 μm
Pitch	0.21 arcsec	±0.30 arcsec	0.26 arcsec	±200 arcsec
Yaw	0.16 arcsec	±0.17 arcsec	0.26 arcsec	±200 arcsec
Roll	0.45 arcsec	±0.60 arcsec	0.69 arcsec	±680 arcsec

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
