# Peer review of "Simultaneous Measurement of 6DOF Motion Errors of Linear Guides of CNC Machine Tools Using Different Modes"

_sensors, 2020, doi:10.3390/s20123439_

Round 1

Reviewer 1 Report

Dear authors,

You presented a measurement method for measuring geometric error of linear guides of CNC machine tools. In this article, you discussed and analyzed the results obtained via the proposed method in two different operation modes. It is an interesting technique for me. The proposed technique seems useful and has its merits. The explanation of this paper is concise. The appearances of the experimental results are approximately complete. The successful experiment results demonstrate your method is capable of sensing 6-DOF motion errors. However, the optical configuration and the measurement principles of the proposed method are too similar to your previous study, according to Ref. 22, [paper title: A Method for Simultaneously Measuring 6DOF Geometric Motion Errors of Linear and Rotary Axes Using Lasers]. Therefore, I do suspect the intrinsic value of this paper is not high enough to publish in this journal unless you could provide more attractive or outstanding experimental data for proofing the creativity of your method.

My recommendation is to reject this article. There are few comments for you to clarify for improving. Please see below for specific comments.

  1. As shown in Table 2, you only present the measurement results obtained by your proposed method (operation mode: 1) and the commercial interferometer (XL-80), comparison with the commercial laser interferometer (XL-80) would be useful for the reader to remove stage errors present in both signals, while illustrating the differences between your proposed method and a well-established technique. Therefore, please also add the measurement results obtained by the commercial interferometer (XL-80) in figures 6 and 7
  2. What about the tolerance ranges for your method?
  3. The proposed technique is based on heterodyne detection configuration, therefore, what is the dominant error source of the existing setup, and what is the solution for improvement and what are predicted results?
  4. The measurement range discussion is needed.
  5. Please discuss the measurement resolutions in each detection axis.

Reviewer 2 Report

I personally enjoyed reading this article very much. It deals with an important aspect in industrial applications, namely the accuracy of CNC machines. 

The paper presents an approach for measuring the 6 DOF simultaneous measurement for geometric error of linear guides in two different modes: (1) the measurement head is fixed and the target mirror moves and respectively (2) the target mirror is fixed and the measurement head moves.

Both the theoretical aspects, the design of the device and the experimental measurements are clearly presented.

It is very rare for me to make such a decision, but I will state that the article can be published as it is. To my knowledge the information is very well presented, is valuable and all the theoretical data is supported by experimental findings. 

Thus, I will end my review by congratulating the authors for their work. 

Reviewer 3 Report

This paper analyzes the simultaneous measurement model of the 6DOF geometric motion error of the linear guide of a CNC machine tool under two different measurement modes. The results show that under two different measurement modes, the straightness error measurement is significantly different, and the positioning error and angle error measurement models are the same. Therefore, in order to ensure the measurement accuracy of the straightness errors in two different measurement modes, different measurement models need to be selected according to the different measurement modes. A series of experiments were built to validate their methods and the accuracy looks quite good. The work is interesting. It has big contributions for precise measurement. Therefore, I recommend it for publication in Sensors with minor revisions.

Q1. For lines 162 and 193, “L” presents different physical meanings. Please revise it.

Q2. There are some typo errors in this paper. For example, line 230, “23.51±%” is strange. Lines 247 and 248,  “5.76um and 7.69um,” should be revised as “5.76 μm and 7.69 μm ”

Q3. For Figs. 3 and 7, I suggest the authors should revise them. For example, The figures and symbols are too small. The authors should label (a), (b), (c)……in these figures.

Reviewer 4 Report

The paper discussed about an interesting topic. However, some parts are not understandable.

  1. The authors say that the system was compared with a laser interferometer (XL80). However, there are no compared results shown in the manuscript. Please show the results.
  2. The authors say that the measured straightness errors depends on the measurement mode in section 4.2. However, the authors also say that the measured results are identical in section 4.3. The reviewer cannot understand what happened between the sections. 
  3.  The word "6DOF motion geometric error" should be revised to "6DOF error motions" because the geometric error can be defined as the errors between axie average lines. The meaured results shown in the manuscript show the error motions, not geometric errors.

Reviewer 5 Report

The authors did a good job with this paper, however, I have only one point.

Please, in conclusion, give a comparison with other studies, that will give more confidence to your work.

Thank you

Round 2

Reviewer 1 Report

Dear Authors,

The paper has been revised to conform as closely as possible to my comments and suggestions. The corresponding descriptions have been modified and added into the article. In addition, the feasibility and performance of your proposed method are further demonstrated through the new and successful experiments. All of the reasoning, discussion, and data in the revised manuscript are clear and understandable. The novelty in terms of optics is rather limited, but the application and its demonstration can be useful. Therefore, in our opinion, this article is suitable for publication in this journal. My recommendation is to accept this study.

Reviewer 4 Report

The word has not been changed.
"6DOF motion geometric error" or "6DOF geometric motion error" in the manuscript has to be corrected to "6DOF error motions".
Please see the definitions in ISO 230-1.

Reviewer 5 Report

Excellent paper, my recommendation is to accept it. 
